# Analysis of the Essential Oils of *Chamaemelum fuscatum* (Brot.) Vasc. from Spain as a Contribution to Reinforce Its Ethnobotanical Use

**Marcos Fernández-Cervantes [1], María José Pérez-Alonso [1], José Blanco-Salas [2,\*]** **,**
**Ana Cristina Soria [3] and Trinidad Ruiz-Téllez [2]**

[1] Department of Biodiversity, Ecology and Evolution. Universidad Complutense, 28071 Madrid, Spain
[2] Department of Vegetal Biology, Ecology and Earth Science, Faculty of Sciences, University of Extremadura, 06071 Badajoz, Spain
[3] Instituto de Química Orgánica General (CSIC), Juan de la Cierva, 3, 28006 Madrid, Spain
[\*] Correspondence: blanco_salas@unex.es; Tel.: +34-9242-89300

**Abstract:** *Chamaemelum fuscatum* (Brot.) Vasc. is a south west Iberian chamomile that has been traditionally used as folk medicine in its natural distribution area but currently it is underestimated regarding its biological activities. For this reason, it is proposed in this paper to get insight into the scientific validation of the traditional knowledge of this plant with the aim of taking advantage of its anti-inflammatory, gastroprotective and antinociceptive activities, among others. To this aim, the chemical composition of the essential oil from the whole plant, the flowers and the green parts of this plant has been evaluated by gas chromatography–mass spectrometry (GC–MS). Plant materials were collected in Badajoz (Spain). A total of 61 components including monoterpenoids, sesquiterpenoids and aliphatic esters were identified. (E)-2-Methyl-2-butenyl methacrylate (27.57%–18.53%) and 2-methylallyl isobutyrate (9.79%–7.51%) were the most abundant compounds in the essential oils of flowers and of the whole plant, whereas α-curcumene, trans-pinocarveol, α-bergamotene and pinocarvone were the major terpenoids irrespective of the plant part considered. Certain compounds showing a relative high abundance as isobutyl methacrylate, isoamyl butyrate, α-bergamotene and pinocarvone were identified for the first time in this species. Finally, we have reviewed the bioactivity of several compounds to relate the ethnobotanical use of this plant in Spain with its volatile profile. This work is a preliminary contribution to reinforce the use to this Mediterranean endemic plant as a natural source of bioactives.

**Keywords:** *Chamaemelum fuscatum*; chamomile; essential oil; aliphatic esters; methacrylate; Compositae; Mediterranean

## 1. Introduction

Ethnobotanical studies often bring to light interesting uses to be further validated. This is the case of a little-known west Mediterranean chamomile, *Chamaemelum fuscatum* (Brot.) Vasc., Asteraceae, frequent in south west of the Iberian Peninsula, forming part of terophytic pastures on wet or temporarily flooded substrates, from 100 to 900 m [1,2]. It is an annual and aromatic little daisy (2–7 cm) easy to identify by its involucral bracts and interfloral scales. It flowers from October to May but mainly in the late winter [1,2].

*C. fuscatum* has traditionally been used as a medicinal plant in Iberian rural areas. The flowers of the species are the most frequently used part [2]. Most of the verified uses of this species have been described for its natural distribution area that roughly coincides with the Luso-Extremadurean region (Table 1).

**Table 1.** Traditional uses for *Chamaemelum fuscatum* in Spain.

| Uses Upon IECTB [1] | Reference | Location | Part Used | Formulation | Preparation | Popular Uses |
|---|---|---|---|---|---|---|
| **HUMAN CONSUMPTION** | | | | | | |
| **Non-alcoholic beverage** | [3] | Cáceres | Inflorescence | Infusion | | Drink |
| **ANIMAL FEEDING** | | | | | | |
| **Fodder** | [4] | Jaén | Inflorescence | | | Birdseed |
| **MEDICINAL USE** | | | | | | |
| **Digestive system** | [5] | Badajoz | Inflorescence | Decoction | Mouthwash | Swollen gums |
| **Digestive system** | [3,5] | Cáceres Badajoz | Inflorescence | Infusion | Drink (sweetened with honey or sugar) | Digestive problems |
| **Digestive system** | [3] | Cáceres | Inflorescence | Infusion | Drink (mixed with olive oil) | Indigestion |
| **Digestive system** | [3] | Cáceres | Inflorescence | Infusion | Drink (mixed with olive oil) | Colic pains |
| **Digestive system** | [3,5] | Cáceres Badajoz | Inflorescence | Infusion | Drink | Laxative |
| **Genito-urinary system** | [5] | Badajoz | Inflorescence | Decoction | Washing of affected area | Relieve vaginal itching |
| **Respiratory system** | [3] | Cáceres | Inflorescence | Decoction | Syrup (mixed with lemon and honey) | |
| **Musculature and skeleton** | [6] | Badajoz | Inflorescence | Decoction | Drink | Antirheumatic |
| **Nervous system and mental illness** | [3,6] | Cáceres Badajoz | Inflorescence | Infusion | Drink (in cáceres sweetened with honey or sugar) | Sedative |
| **Sense organs** | [3,5] | Cáceres Badajoz | Inflorescence | Infusion (Cáceres); decoction (Badajoz) | Eye drops | Eye irritation, conjunctivitis |
| **Other infectious and parasitic diseases** | [5] | Badajoz | Inflorescence | Infusion | Drenching of affected area | Herpes ("feve") |
| **VETERINARY USE** | | | | | | |
| **Digestive system** | [7] | Cáceres | Inflorescence | Infusion | Drink (mixed with olive oil, brandy or bicarbonate) | Digestive problems, lack of rumination, accumulation of gases |
| **Musculature and skeleton** | [7] | Badajoz | Green parts and leaves | Decoction | Rubbing | Lameness and inflammation |
| **Skin and subcutaneous tissue** | [7] | Badajoz | Green parts and leaves | Decoction | Rubbing | Wounds |

**Table 1.** *Cont.*

| Uses Upon IECTB [1] | Reference | Location | Part Used | Formulation | Preparation | Popular Uses |
|---|---|---|---|---|---|---|
| **HUMAN CONSUMPTION** | | | | | | |
| **Non-alcoholic beverage** | [3] | Cáceres | Inflorescence | Infusion | | Drink |
| **ANIMAL FEEDING** | | | | | | |
| **Fodder** | [4] | Jaén | Inflorescence | | | Birdseed |
| **INDUSTRY AND CRAFTS** | | | | | | |
| **Cosmetics, perfumes and cleaning products** | [3] | Cáceres | Whole plant | Decoction | | Dyed blond hair |
| **Clothing and personal adornment** | [3] | Cáceres | Inflorescence | | | Personal adornment |

[1] IECTB: Spanish Inventory of Traditional Knowledge Concerning Biodiversity.

The medicinal applications of *C. fuscatum* for digestive pathologies, as antiseptic or anti-inflammatory remedy for external use were those traditionally in force [2]. However, and despite its multiple bioactivities, the current use of this species is less than that of other chamomiles. Its harvesting for marketing today is insignificant and it remains someway underestimated [2].

Regarding the chemical composition of *C. fuscatum*, it started to be studied in the eighties at Salamanca University (Spain). De Pascual Teresa et al. [8] analyzed the hexane extract of the aerial parts of the plant, and first identified the methacrylic esters of 2-methyl-2-*(E)*-butenol, 2-hydroxy-2-methyl-3-butenol and 2-hydroxy-2-methyl-3-oxobutanol by spectral measurements and analysis of the corresponding standards. After that, these authors used spectroscopic methods and chemical transformations to elucidate the structures of different eudesmanolides in the chloroform extract of the aerial parts of this plant [9], and two years later, they reported four new eudesmanolides in the same extract and a new aliphatic ester (2-methylene-3-oxobutyl methacrylate) in the essential oil isolated from flowers [10]. Since then, and as far as we know, no information has been published on the chemical profile of *C. fuscatum*. Therefore, it is proposed as the main objective of this work to undertake a detailed characterization by gas chromatography–mass spectrometry (GC–MS) of *C. fuscatum* essential oils, whose composition is still deeply unknown. Moreover, a comprehensive bibliographic review of the bioactivity of the compounds here determined has been done to relate the ethnobotanical use of this plant in Spain with its volatile profile. It is expected that getting insight into the scientific validation of the traditional use of this plant is a contribution to confer more value to *C. fuscatum* as a new source of bioactives and, therefore, it also contributes to guarantee its preservation.

## 2. Materials and Methods

### 2.1. Bibliographic Prospection

After an exhaustive review of the Spanish ethnobotanical literature following the methodology of the Spanish Inventory of Traditional Knowledge Concerning Biodiversity (IECTB) [11], the traditional uses of this species in Spain was summarized in this paper. This information was complemented with a bibliographic review aimed to get insight into the chemical composition and pharmacological activities of this species. The following resources were accessed: Academic Search Complete, Agricola, Agris, Biosis, CAB Abstracts, Cochrane, Cybertesis, Dialnet, Directory of Open Access Journals, Embase, Espacenet, Google Academics, Google Patents, Medline, PubMed, Science Direct, Scopus, Teseo and Web of Science by the Institute for Scientific Information (ISI).

As for the systematic review of pharmacologically active components of *C. fuscatum*, the general procedures of Prisma 2009 Flow Diagram [12] were followed. Keywords selected as search terms were the compounds listed in Table 2 with a concentration higher than 1% and "activity" or "pharmacol*" or "biological activity".

### 2.2. Plant Material

*C. fuscatum* samples were collected in March 2017 in Cerro del Viento (Badajoz, Spain) at the following coordinates: 29SPD70, 38°51′46.2″ N, 6°58′35.15″ W. A voucher specimen with reference number HSS 68,118 was lodged at the Herbarium of the Research Center Finca La Orden, CicyTex, Junta de Extremadura, Badajoz (Spain). Samples collected at their flowering stage were air-dried and stored in the dark at 10–15 °C until distillation.

### 2.3. Isolation of C. fuscatum Essential Oil

The essential oils from the different parts (flowers (F), whole plant (WP) and green parts (GP), which includes a mixture of leaves and stems), of *C. fuscatum* were isolated by hydrodistillation with cohobation for 8 h, using a Clevenger modified apparatus, according to the method recommended in the Spanish Pharmacopeia. The oils were dried over anhydrous magnesium sulphate and stored at 4

°C in the dark until analysis. The extraction yield (%) was calculated as the amount of essential oil (in g) extracted by hydrodistillation from 100 g of dry plant.

### 2.4. Gas Chromatography–Mass Spectrometry (GC–MS) Analysis

GC–MS analyses were carried out on a 7890A gas chromatograph coupled to a 5975C quadrupole mass detector (both from Agilent Technologies, Palo Alto, CA, USA), using He at ~1 mL min$^{-1}$ as carrier gas. Injections were carried out in split mode (1:20) at 250 °C. Separations were performed using a Zebron 5% phenylmethyl silicone column (30 m × 0.25 mm, 0.25 μm film thickness) from Phenomenex (Madrid, Spain). Oven temperature program was raised from 70 °C (0.5 min) to 290 °C (30 min) at 6 °C min$^{-1}$. Mass spectra were recorded in electron impact (EI) mode at 70 eV, scanning the 35–450 *m/z* range. Interface and source temperature were set at 280 °C and 230 °C, respectively. Acquisition was done using HP ChemStation software (Agilent Technologies).

Qualitative analysis was based on the comparison of experimental mass spectra with data from the Wiley mass spectral library [13] and was confirmed, when possible, by using linear retention indices ($I^T$) [14,15]. Semiquantitative data (percentage of total volatiles) were directly calculated from peak areas of total ion current (TIC) profiles.

**Table 2.** Essential oil composition of the different parts of *C. fuscatum* (F: Flowers, S: Green parts, WP: Whole plant). Class of identified compound is marked with a superscript ([a]: Aliphatic esters, [b]: Monoterpenes, [c]: Sesquiterpenes, [d]: Others). [e]: Retention indices on the DB5 column were taken from [14] except those marked with '*', which were taken from [15]. -: Not found; n.i.: Non identified. Unidentified components less than 0.5% were not included in this table.

| Peak | Compound | $I^T$ (exp.) | $I^T$ (lit.) [e] | Percent Composition (%) | | |
| | | | | F | WP | GP |
|---|---|---|---|---|---|---|
| 1 | 2-methylpropyl isobutyrate [a] | 895 | 892 | 1.54 | 0.54 | - |
| 2 | 2-methylallyl isobutyrate [a] | 927 | - | 9.79 | 7.51 | - |
| 3 | α-pinene [b] | 939 | 932 | 0.71 | 1.11 | - |
| 4 | isobutyl methacrylate [a] | 960 | - | 2.38 | 2.14 | - |
| 5 | isobutyl 2-methylbutyrate [a] | 1004 | 1002 * | 0.20 | 0.16 | - |
| 6 | 2-methylbutyl isobutyrate [a] | 1015 | 1014 * | 2.03 | 1.66 | - |
| 7 | limonene [b] | 1031 | 1031 | 0.34 | 0.10 | - |
| 8 | 1,8-cineol [b] | 1033 | 1032 | 0.50 | 0.30 | - |
| 9 | γ-terpinene [b] | 1060 | 1054 | 0.10 | 0.40 | - |
| 10 | isoamyl butyrate [a] | 1060 | 1060 | 3.60 | 1.64 | - |
| 11 | (E)-2-methyl-2-butenyl methacrylate [a] | 1087 | - | 27.57 | 18.53 | 0.73 |
| 12 | 3-methylbutyl-2-methyl-butyrate [a] | 1100 | 1100 | 0.27 | 0.15 | - |
| 13 | 2-methylbutyl-2-methyl-butyrate [a] | 1106 | 1103 | 0.30 | 0.17 | - |
| 14 | 3-methyl-3-butenyl isovalerate [a] | 1118 | 1116 * | 1.22 | 1.10 | - |
| 15 | α-canfolenal [b] | 1119 | 1122 | 0.33 | 0.15 | - |
| 16 | trans-pinocarveol [b] | 1139 | 1135 | 5.14 | 2.90 | 0.20 |
| 17 | camphor [b] | 1143 | 1141 | 0.20 | 0.25 | - |
| 18 | pinocarvone [b] | 1162 | 1160 | 4.39 | 2.62 | 0.45 |
| 19 | 3-pinanone [b] | 1173 | 1172 | 0.43 | 0.28 | - |
| 20 | terpinen-4-ol [b] | 1179 | 1174 | 0.20 | 0.20 | - |
| 21 | myrtenol [b] | 1194 | 1194 | 1.17 | 0.89 | - |
| 22 | amyl tiglate [a] | 1229 | 1126 * | 0.82 | 0.44 | - |
| 23 | (-)-carvone [b] | 1242 | 1239 | 0.53 | 0.35 | - |
| 24 | nonanoic acid [d] | 1280 | 1267 | 0.16 | 0.10 | - |
| 25 | cis-myrtenal [b] | 1289 | 1295 | 0.19 | 0.15 | - |

**Table 2.** *Cont.*

| | | | | Percent Composition (%) | | |
|---|---|---|---|---|---|---|
| Peak | Compound | $I^T$ (*exp.*) | $I^T$ (*lit.*) [e] | F | WP | GP |
| 26 | geranyl formate [b] | 1300 | 1298 | 0.35 | - | - |
| 27 | (*E,E*)-2,4-decadienal[d] | 1318 | 1316 | 0.38 | - | - |
| 28 | myrtenyl acetate [b] | 1326 | 1326 | 0.20 | 0.20 | - |
| 29 | benzyl methacrylate [a] | 1357 | - | 0.39 | 0.10 | - |
| 30 | decanoic acid [d] | 1369 | 1364 | 0.56 | 0.76 | - |
| 31 | (*E*)-*β*-damascenone [b] | 1380 | 1384 | 0.19 | 0.49 | - |
| 32 | phenylethyl isobutyrate [a] | 1396 | 1396 * | 0.35 | 0.19 | - |
| 33 | *trans*-caryophyllene [c] | 1404 | 1408 | 0.33 | 0.32 | - |
| 34 | *α*-bergamotene [c] | 1438 | 1412 | 5.08 | 4.94 | 2.18 |
| 35 | *α*-curcumene [c] | 1483 | 1480 | 9.21 | 8.06 | 4.69 |
| 36 | *α*-muurolene [c] | 1499 | 1500 | 0.23 | 0.27 | 0.62 |
| 37 | (*E,E*)-*α*-farnesene [c] | 1509 | 1505 | 0.15 | - | - |
| 38 | *γ*-cadinene [c] | 1513 | 1513 | 0.50 | 0.86 | 0.31 |
| 39 | (*Z*)-*γ*-bisabolene [c] | 1515 | 1515 | 0.28 | 0.36 | - |
| 40 | *δ*-cadinene [c] | 1524 | 1523 | 1.39 | 1.73 | 1.30 |
| 41 | cadina-1,4-diene [c] | 1532 | 1534 | 0.39 | 0.20 | - |
| 42 | (*Z*)-nerolidol [c] | 1534 | 1532 | 0.60 | 0.43 | 0.30 |
| 43 | *α*-cadinene [c] | 1538 | 1538 | 0.38 | 0.20 | - |
| 44 | neryl isovalerate [c] | 1576 | 1583 | 1.32 | 2.98 | 6.80 |
| 45 | spathulenol [c] | 1578 | 1578 | 0.84 | 0.64 | 0.83 |
| 46 | caryophyllene oxide [c] | 1581 | 1578 | 0.10 | 0.27 | - |
| 47 | guaiol [c] | 1595 | 1600 | 0.77 | 0.76 | 0.89 |
| 48 | isoamyl nerolate [c] | 1601 | 1602 | 0.69 | 0.23 | 1.11 |
| 49 | 1,10-di-*epi*-cubenol[c] | 1614 | 1619 | 0.33 | 0.28 | 0.72 |
| 50 | *γ*-eudesmol [c] | 1630 | 1632 | 0.10 | 0.44 | 1.77 |
| 51 | himachalol [c] | 1647 | 1653 | 0.10 | 0.38 | 0.96 |
| 52 | *α*-cadinol [c] | 1652 | 1654 | 0.10 | 0.28 | 0.26 |
| 53 | *β*-eudesmol [c] | 1659 | 1650 | 0.10 | 0.10 | - |
| 54 | *α*-bisabolol [c] | 1682 | 1685 | 0.39 | 0.71 | 0.72 |
| 55 | xanthorrhizol [c] | 1751 | 1753 | - | - | 0.21 |
| 56 | pentadecanol [c] | 1778 | 1774 | - | - | 0.75 |
| 57 | *β*-bisabolenol [c] | 1786 | 1789 | 0.39 | 0.28 | 0.75 |
| 58 | 1-octadecene [d] | 1793 | 1790 | 0.52 | 0.41 | 1.06 |
| 59 | hexadecanol [d] | 1879 | 1875 | - | 0.20 | 1.24 |
| 60 | hexadecanoic acid [d] | 1972 | 1960 | 2.49 | 10.74 | 23.89 |
| 61 | octadecanoic acid [d] | 2180 | 2180 | 0.75 | 8.65 | 18.68 |
| | Aliphatic esters ([a]) | | | 50.46 | 34.33 | 0.73 |
| | Monoterpenoids ([b]) | | | 14.97 | 10.39 | 0.65 |
| | Sesquiterpenoids ([c]) | | | 23.77 | 24.72 | 24.42 |
| | Others ([d]) | | | 4.86 | 20.86 | 45.62 |
| | % identified | | | 94.06 | 90.30 | 71.42 |

## 3. Results

### 3.1. Yield

*C. fuscatum* gave yellow essential oils with yields of 0.40%, 0.11% and 0.25% for flowers (F), green parts (GP) and whole plant (WP), respectively. The volatiles in the essential oils here analyzed (94.06% F, 71.42% S, 90.30% WP) were identified based on their mass spectra and chromatographic retention data. Table 2 lists these compounds according to their linear retention indices ($I^T$).

## 3.2. Chemical Composition of C. fuscatum Essential Oils

Aliphatic esters were the predominant class of compounds identified in the essential oils of flowers and the whole plant (50.46% and 34.33%, respectively), followed by sesquiterpenoids (23.77%–24.72%) and monoterpenoids (10.39%–14.97%). There was a noticeable absence of aliphatic esters in the green parts of this species, with (E)-2-methyl-2-butenyl methacrylate representing only 0.73% of the essential oil of green parts.

We have identified for the first time in this species new aliphatic esters with relative abundances higher than 1% and showing very similar structures to those of compounds previously reported in [8–10]. It is also worth noting that several new aliphatic esters with relative abundances higher than 1% and showing very similar structures to compounds previously reported in [8–10] were also identified for the first time in this paper (see Figure 1).

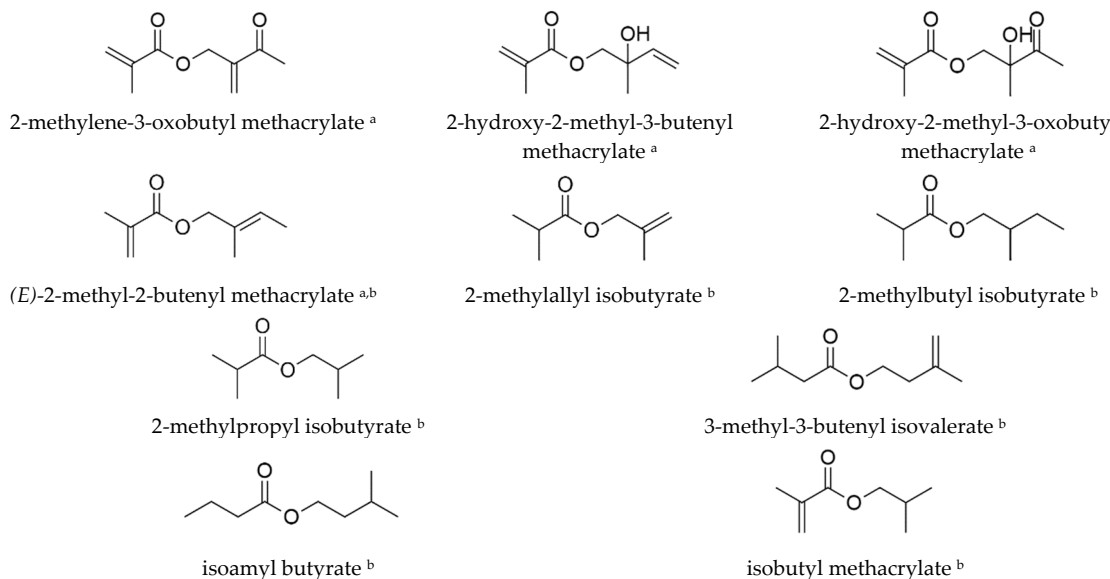

**Figure 1.** Chemical structures of selected aliphatic esters identified in *C. fuscatum*. [a:] Aliphatic esters previously reported by De Pascual et al. [8–10], [b:] Aliphatic esters with relative abundances higher than 1% identified in this work.

A total of 42 terpenoids have also been identified in the essential oils of the aerial parts of *C. fuscatum*. α-Curcumene (9.21%), trans-pinocarveol (5.14%), α-bergamotene (5.08%) and pinocarvone (4.39%) represent a high percentage of the terpenoid fraction of the flower essential oil. Neryl isovalerate was detected in green parts in percentages higher than those found for F and WP essential oils (6.80% vs. 1.32% and 2.98%, respectively). Sesquiterpenoids were the major terpenoid class (24.42%) in the essential oil of green parts, with a noticeable contribution (45.62%) of other compounds mainly long chain fatty acids as hexadecanoic and octadecanoic acids.

## 3.3. Review of the Pharmacological Activities of C. fuscatum Essential Oil Compounds

Although most of essential oils isolated from *C. fuscatum* showed high relative contents of methacrylic and butyric esters, the medicinal value of this kind of compounds is still deeply unknown. (E)-2-methyl-2-butenyl methacrylate has been reported to have an intense odor [10]; however, as far as we know, no information is available on its effects over animal or human physiology.

On the other hand, the main terpenoids with reported pharmacological activities present in the essential oils of *C. fuscatum* are summarized in Table 3. Antinociceptive (α-curcumene, myrtenol), anti-inflammatory (α-curcumene, myrtenol, spathulenol, α-bisabolol, α-bergamotene), anxiolytic ((Z)-nerolidol), antimicrobial (*trans*-pinocarveol, α-pinene, γ-terpinene, δ-cadinene) and even anticarcinogen (spathulenol) compounds have been detected.

**Table 3.** Review of the main terpenoids with reported pharmacological activities present in *C. fuscatum* essential oils.

| Compound | Reported Pharmacological Activities | References |
|---|---|---|
| α-Pinene | Antibacterial, antifungal | [16] |
| | Hypotensive | [17] |
| | Gastroprotective | [18] |
| Limonene | Anti-inflammatory, antioxidant | [19] |
| | Antimicrobial | [20] |
| 1,8-Cineole | Gastroprotective | [21] |
| | Anti-inflammatory | [22] |
| γ-Terpinene | Antioxidant | [23] |
| | Antibacterial | [24] |
| *Trans*-pinocarveol | Antibacterial | [25] |
| Camphor | Analgesic, anti-inflammatory | [26] |
| Myrtenol | Anti-inflammatory, antinociceptive | [27] |
| *(E)-β*-Damascenone | Antispasmodic | [28] |
| α-Bergamotene | Anti-inflammatory | [29] |
| α-Curcumene | Stomachic, antinociceptive, anti-ulcer | [30] |
| | Anti-inflammatory | [31] |
| δ-Cadinene | Anti-inflammatory | [29] |
| | Antibacterial, antioxidant | [32] |
| *(Z)*-Nerolidol | Antinociceptive | [33] |
| | Sedative | [34] |
| Spathulenol | Anti-inflammatory, antinociceptive, anticarcinogen | [35] |
| α-Bisabolol | Anti-inflammatory | [36] |
| | Digestive | [37] |

## 4. Discussion

The medicinal properties of chamomiles are usually attributed to their essential oil components. *C. fuscatum* mainly concentrates its essential oil in the flowers (0.40%), and in smaller proportions in green parts (0.11%). For this reason, flowers are the part of the plant generally used as folk medicine. As compared to other chamomile species, the yield of *C. fuscatum* flower essential oil (0.40%) was similar to that of other relative species. Roman chamomile (*Chamaemelum nobile)*, its closest relative, has been described to usually yield around 0.70% of essential oil and a yield range between 1.90%–0.24% has been reported for German chamomile (*Matricaria recutita*) [38].

As mentioned, aliphatic esters had been previously reported in *C. fuscatum* [8–10] and also in related chamomile species as *Chamaemelum nobile* and in several species of the *Anthemis* genus [39]. *(E)*-2-methyl-2-butenyl methacrylate had been described in the hexane extract of the aerial parts by De Pascual et al. [8], together with other methacrylates (2-hydroxy-2-methyl-3-butenyl methacrylate and 2-hydroxy-2-methyl-3-oxobutyl methacrylate) that have not been detected here. 2-Methylene-3-oxobutyl methacrylate, a methacrylic ester previously isolated from the essential oil of *C. fuscatum* [10] was not found either in the essential oils here analyzed. However, we have identified in a relative high abundance (>1%) the esters isobutyl methacrylate and isoamyl butyrate, which had not been cited in previous literature of this species [8–10]. These differences could be attributed to the different extraction procedures with respect to [8–10], but also to the different environments since the population here analyzed was collected in Badajoz (Spain) and the samples analyzed in the cited studies were from Cáceres (Spain).

The pharmacological activities of the methacrylic and butyric esters are still unknown. As these compounds are present in high proportions in the volatile fraction of common medicinal species such as chamomiles, investigation on this topic would be worthy of consideration. In addition, it would be interesting to investigate if the structural similitudes among these classes of compounds could support that they act over the same biological paths.

From the 42 terpenoids identified in the aerial parts of *C. fuscatum*, only α-curcumene, neryl isovalerate and trans-pinocarveol had been previously reported [8–10]. Moreover, several terpenoids with a relative high abundance such as pinocarvone and α-bergamotene have been identified for the first time in this species. Together with aliphatic esters, *trans*-pinocarveol and pinocarvone have been reported as major compounds of *C. nobile* essential oils [39].

Despite its hydrophobic nature, terpenoids have also been detected in chamomile teas [40] and, therefore, they could play a role in the medicinal value of chamomile tea prepared from *C. fuscatum* flowers. In this work, we have summarized the reported pharmacological activities of several terpenoids that occur in the essential oils of this species. These activities could be related with the traditional uses of this species, which is also interesting in the context of validating its medicinal value. As an example, the presence of sesquiterpenoids with stomachic and gastroprotective activity such as α-curcumene, α-bisabolol and 1,8-cineole could support the fact that *C. fuscatum* has been traditionally used as digestive. In fact, α-bisabolol has been previously related to the digestive action of German chamomile [37].

As previously mentioned, the composition of *C. fuscatum* essential oils showed similitudes with that of essential oils isolated from *C. nobile*. However, there are chemical differences between both species that could result in different physiological responses. Therefore, comparative studies regarding bioactivity must be carried out before *C. fuscatum* can be raised as an interesting alternative to this plant. In contrast, the volatile profile of essential oils here analyzed differs drastically from that of German chamomile, since this plant is especially rich in bisaboloids (> 50% according to [38]), which are found in low relative contents in the essential oil obtained from *C. fuscatum* flowers. Nevertheless, the presence of α-bisabolol, one of the major constituents of German chamomile, in *C. fuscatum* essential oils here analyzed make worthy of investigation the role of this compound in the digestive action of both species.

## 5. Conclusions

In this work, a comprehensive evaluation regarding chemical composition and ethnobotanical use of a scarcely studied species, *C. fuscatum*, was carried out. In addition to a number of different monoterpenoids, sesquiterpenoids, etc., methacrylates and butyrates were determined as major components of the essential oils of this plant, being (*E*)-2-methyl-2-butenyl methacrylate the compound showing the highest percent content in both flower and whole plant essential oils. Moreover, several compounds with high relative abundances such as isobutyl methacrylate, isoamyl butyrate, α-bergamotene and pinocarvone were also identified for the first time in this plant. The published literature on the bioactivity of these compounds has been correlated with GC–MS data to support the popular knowledge and ethnobotanical use of this Mediterranean chamomile. Despite studies regarding the evaluation of the effect of different factors such as cultivation region, climate, extraction method, etc. will be further addressed, results of this preliminary study highlighted the potential of this plant as a natural source of bioactives of application in the pharmacological, cosmetic and food industries, among others.

**Author Contributions:** Conceptualization, T.R.-T.; Methodology, M.J.P.-A. and J.B.-S.; Validation, M.J.P.-A.; Formal Analysis, M.F.-C. and M.J.P.-A.; Investigation, M.F.-C., A.C.S., M.J.P.-A.; Data Curation, J.B.-S and M.F.-C.; Writing-Original Draft Preparation, T.R.-T.; Writing-Review and Editing, J.B.-S. and T.R.-T.; Visualization, J.B.-S.; Supervision, M.J.P.-A.; Project Administration, T.R.-T.; Funding Acquisition, M.J.P.A and T.R.-T.

**Funding:** This research was partially funded by IB16003 project financed by the Junta of Extremadura (Spain) and the European Regional Development Fund.

**Acknowledgments:** To Manuel Pardo de Santayana and his team (Inventario Español de Conocimientos Tradicionales Relativos a la Biodiversidad, IECTB) for helping with the bibliographic ethnobotanical review.

**Conflicts of Interest:** The authors declare no conflict of interest. The founding sponsors had no role in the design of the study; in the collection, analyses, or interpretation of data; in the writing of the manuscript, and in the decision to publish the results.

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
