# Peer review of "Analysis of the Essential Oils of Chamaemelum fuscatum (Brot.) Vasc. from Spain as a Contribution to Reinforce Its Ethnobotanical Use"

_forests, doi:10.3390/f10070539_

Round 1

Reviewer 1 Report

This paper describes the chemical constituents of the essential oils of Chamaemelum fuscatum in a satisfactory way. The work is interesting, informative and well-written. I have some observations that are listed in the following lines.

1.       Line 92: What was the distillation rate?

2.       Line 95: If I understand this statement correctly, the yield was calculated as (%) by dividing amount of essential oil (in mL) from 100 grams of dry plant (which gives mL/g). It should have been calculated by using the weight of the essential oil in grams per weight of dry plant and multiplied by 100.

3.       Line 108: did the authors run a GC-FID analysis of the studied oils? Since this is the first time to report the oil composition, could the authors carry out this GC-FID analysis with as much calibration as possible against authentic standards, for the quantitative data?

4.       The experimental design included a single extraction and analysis of each oil but, to be correct, the entire experiment should have been replicated at least once (desirably more times), with another batch of the plant part used.

5.       Table 2: The % of identified compounds from the stem oil is pretty low (71.42%).

6.       Line 205: this sentence is contradicting itself. How can the plant be scarcely studied but wide variety of bioactivities carried out?

Author Response

This paper describes the chemical constituents of the essential oils of Chamaemelum fuscatum in a satisfactory way. The work is interesting, informative and well-written. Thank you so much. I have some observations that are listed in the following lines.

1. Line 92: What was the distillation rate?

It was about 20 drops by min during the first 3h and then slowly decreased. 8h, however, were necessary in order to carry out an exhaustive extraction of the less volatile compounds.

2. Line 95: If I understand this statement correctly, the yield was calculated as (%) by dividing amount of essential oil (in mL) from 100 grams of dry plant (which gives mL/g). It should have been calculated by using the weight of the essential oil in grams per weight of dry plant and multiplied by 100.

Several articles refer the yield as mL/g (v/w), it is very usual in the Essential Oil research. Examples:

- Ben Hsouna, Ben Halima N, Smaoui S, Hamdi N 2017. Citrus lemon essential oil: chemical composition, antioxidant and antimicrobial activities with its preservative effect against Listeria monocytogenes inoculated in minced beef meat. Lipids Health Dis. 16(1):146-157.

- Verma, R.S., Padalia, R.C., Chauhan, A. 2016. Chemical composition of essential oil and rosewater extract of Himalayan Musk Rose (Rosa brunonii Lindl.) from Kumaon region of western Himalaya. Journal of Essential Oil Research, 28, 4, 332–338.

- Arrebola, M.L., Navarro, M.C., Jiménez, J., Ocaña, F.A. 1994. Yield and composition of the essential oil of Thymus serpylloides subsp. serpylloides. Phytochemistry, 36, 1, 67–72.

However, changes have been applied and yield is now expressed as w/w

3. Line 108: did the authors run a GC-FID analysis of the studied oils? Since this is the first time to report the oil composition, could the authors carry out this GC-FID analysis with as much calibration as possible against authentic standards, for the quantitative data?

We’re focusing on obtaining quantitative data more exhaustively in our next studies. This is a preliminary study in which the aromatic composition was analysed by GC-MS for the first time.

4. The experimental design included a single extraction and analysis of each oil but, to be correct, the entire experiment should have been replicated at least once (desirably more times), with another batch of the plant part used.

You are right and it would have been desirable to replicate the experiments as much as possible. But even though, for a firs approximation and description of new compounds, there is plenty of published papers on essential oil analysis that perform the study with just one sample. The collection of the material is very important in these cases, because if the method proposal contains just 1 extraction, the plant material oughts to come from different individuals.

5. Table 2: The % of identified compounds from the stem oil is pretty low (71.42%).

We specially focused on the flowerheads  (>90%) since they’re the most important part of this species according to folk medicine. The compounds of the stems were rare sesquiterpenoids (we suppose considering their mass spectra) with complex structures whose identification was difficult by GC-MS. In our next project we’re using HPLC too in order to obtain more data related to the green parts of the plant.

6. Line 205: this sentence is contradicting itself. How can the plant be scarcely studied but wide variety of bioactivities carried out?

That’s true. That sentence was wrong. Changes have been applied.

Reviewer 2 Report

In the present study the others investigate chemical composition the Essential Oils of Chamaemelum fuscatum (Brot.) Vasc. growing in Cerro del Viento (Badajoz, Spain). This is a very interesting paper, the other hand this work presents some corrections according to the following comments.

Comments to Authors:

1-      Page 1 lin 27: It is necessary to specify the origin of your plant in the title and the abstract.

2-      Page 3 lin 91: Why you have not use the leaves and roots. You speak of the whole plant what does it mean: did you use the complete plant or mixed flowers (F) and stems (S) only?

3-      Page 3 lin 94: It will be better for you to weigh the amount of the essential oil in (mg) and not in (mL) produced per quantity of plant mass to have a correct extraction yield (%).

4-      In the discussion you must explain the presence of certain compounds identified for the first time in this species, especially which your plant comes from a geographical site (Badajoz, Spain) already study in the past in some studies.

5-      In spite of its import, the work lacks solid and consistent results, I think, it is important to move to a more in-depth biological study to value your chemical discovery in terms of new products and see their effects in this context compared to the literature.

Author Response

In the present study the others investigate chemical composition the Essential Oils of Chamaemelum fuscatum (Brot.) Vasc. growing in Cerro del Viento (Badajoz, Spain). This is a very interesting paper (thank you), the other hand this work presents some corrections according to the following comments.

Comments to Authors:

1- Page 1 lin 27: It is necessary to specify the origin of your plant in the title and the abstract.

OK. Done.

2- Page 3 lin 91: Why you have not use the leaves and roots. You speak of the whole plant what does it mean: did you use the complete plant or mixed flowers (F) and stems (S) only?

Leaves were considered too in the stems. We’ve changing this category to Green Parts (GP). Also, we’ve specified which parts were the ‘green parts’ in the experimental section.

When we analyzed the whole plant we used the complete plant, including the roots.

3- Page 3 lin 94: It will be better for you to weigh the amount of the essential oil in (mg) and not in (mL) produced per quantity of plant mass to have a correct extraction yield (%).

Several articles refer the yield as mL/g (v/w), it is very usual in the Essential Oil research. Examples:

- Ben Hsouna, Ben Halima N, Smaoui S, Hamdi N 2017. Citrus lemon essential oil: chemical composition, antioxidant and antimicrobial activities with its preservative effect against Listeria monocytogenes inoculated in minced beef meat. Lipids Health Dis. 16(1):146-157.

- Verma, R.S., Padalia, R.C., Chauhan, A. 2016. Chemical composition of essential oil and rosewater extract of Himalayan Musk Rose (Rosa brunonii Lindl.) from Kumaon region of western Himalaya. Journal of Essential Oil Research, 28, 4, 332–338.

- Arrebola, M.L., Navarro, M.C., Jiménez, J., Ocaña, F.A. 1994. Yield and composition of the essential oil of Thymus serpylloides subsp. serpylloides. Phytochemistry, 36, 1, 67–72.

However, changes have been applied and yield is now expressed as w/w.

4- In the discussion you must explain the presence of certain compounds identified for the first time in this species, especially which your plant comes from a geographical site (Badajoz, Spain) already study in the past in some studies.

Thank you for the observation. We have included this fact in the text.

5- In spite of its import, the work lacks solid and consistent results, I think, it is important to move to a more in-depth biological study to value your chemical discovery in terms of new products and see their effects in this context compared to the literature.

We agree with you, and we present this work as the first one of a series of studies that we are going to carry on in the direction you suggest. One of the authors is preparing his Ph Thesis in this line.

Reviewer 3 Report

The paper "Analysis of the Essential Oils of Chamaemelum fuscatum (Brot.) Vasc. as a Contribution to Reinforce Its Ethnobotanical Use" presents both experimental data and a review of ethnobotanical uses of EO of C. fuscatum. The methods are clearly stated and the results presented with clarity. The discussion benefits from a very good literature search.

Author Response

The paper "Analysis of the Essential Oils of Chamaemelum fuscatum (Brot.) Vasc. as a Contribution to Reinforce Its Ethnobotanical Use" presents both experimental data and a review of ethnobotanical uses of EO of C. fuscatum. The methods are clearly stated and the results presented with clarity. The discussion benefits from a very good literature search.

Thank you so much.

Round 2

Reviewer 1 Report

The authors have revised the manuscript as recommended. I think it is ready for publication.

Reviewer 2 Report

The others responded well to the different suggestions, I recommend the publication of this present work in the form of short communication.